# Pharmacological Activation of YAP/TAZ by Targeting LATS1/2 Enhances Periodontal Tissue Regeneration in a Murine Model

**DOI:** 10.3390/ijms24020970

**Published:** 2023-01-04

**Authors:** Akiko Sato, Shigeki Suzuki, Hang Yuan, Rahmad Rifqi Fahreza, Xiuting Wang, Eiji Nemoto, Masahiro Saito, Satoru Yamada

**Affiliations:** 1Department of Periodontology and Endodontology, Tohoku University Graduate School of Dentistry, Sendai 980-8575, Japan; 2Department of Restorative Dentistry, Tohoku University Graduate School of Dentistry, Sendai 980-8575, Japan

**Keywords:** periodontal tissue regeneration, YAP/TAZ, periodontal ligament fibroblasts, LATS1/2, MAP4K4, DPPA

## Abstract

Due to their multi-differentiation potential, periodontal ligament fibroblasts (PDLF) play pivotal roles in periodontal tissue regeneration in vivo. Several in vitro studies have suggested that PDLFs can transmit mechanical stress into favorable basic cellular functions. However, the application of mechanical force for periodontal regeneration therapy is not expected to exhibit an effective prognosis since mechanical forces, such as traumatic occlusion, also exacerbate periodontal tissue degeneration and loss. Herein, we established a standardized murine periodontal regeneration model and evaluated the regeneration process associated with cementum remodeling. By administering a kinase inhibitor of YAP/TAZ suppressor molecules, such as large tumor suppressor homolog 1/2 (LATS1/2), we found that the activation of YAP/TAZ, a key downstream effector of mechanical signals, accelerated periodontal tissue regeneration due to the activation of PDLF cell proliferation. Mechanistically, among six kinds of MAP4Ks previously reported as upstream kinases that suppressed YAP/TAZ transcriptional activity through LATS1/2 in various types of cells, MAP4K4 was identified as the predominant MAP4K in PDLF and contributed to cell proliferation and differentiation depending on its kinase activity. Ultimately, pharmacological activation of YAP/TAZ by inhibiting upstream inhibitory kinase in PDLFs is a valuable strategy for improving the clinical outcomes of periodontal regeneration therapies.

## 1. Introduction

Periodontitis is a multi-factorial disease accompanied by irreversible destruction of periodontal tissue such as alveolar bone, cementum, periodontal ligament (PDL), and gingival tissue. The disease is developed by a local pathogenic bacterial infection and involves the host’s protective inflammatory reactions, which release cytokines and enzymes that induce the breakdown of periodontal connective tissue and bone [1]. Generally, chronic inflammation causes the onset and progression of various diseases including not only periodontitis but other diseases such as obesity, diabetes, cardiovascular disease, and cancer [2,3,4]. Prolonged inflammation in bone tissues increases the number and activity of osteoclasts and induces osteoblast malfunction and apoptosis, thereby losing bone tissue regeneration capability [5]. Periodontal treatment, including mechanical and chemical removal of infection sources, oral hygiene control, and lifestyle improvement, increases host resistance and prevents further inflammatory tissue destruction. However, even after inflammation settles down in periodontal tissue, the alveolar bone loss remains almost unchanged and only the nominal cementum layer or PDL tissue is regenerated. Therefore, root exposure, cosmetic issue, increased caries risk, and recurrence risk of periodontitis are maintained [6]. Several periodontal regeneration therapies, such as making space for mesenchymal cells to migrate by guided tissue regeneration (GTR) and topical application of Emdogain gel or recombinant FGF-2 gel, have been clinically applied [7,8]. These biocompatible treatments partially achieve physiological periodontal tissue regeneration of vertical alveolar bone defects and defects limited to the furcation area. However, the ultimate goal of periodontal regeneration therapy is complete physiological periodontal tissue regeneration, even for cases involving widely-spread horizontal bone defects, which are common phenotypes in periodontitis patients. One of the major challenges to the effective application of current regeneration therapies for horizontal bone defects is the low population of PDL cells compared with the area of the bone defect [9]. Therefore, new approaches that maximize the proliferation of PDL cells are necessary.

Mechanotransduction is a basic cellular mechanism used in many types of cells, by which mechanical stress from the external environment, such as changes in hardness, stiffness, and shape of the extracellular matrix, is transmitted into biological signals [10]. For instance, mechanotransduction is evoked under stress in myocardial cells by the beating of the heart, in vascular endothelial cells by blood pressure, in osteocytes and osteoblasts by gravity, and in alveolar macrophages by breathing [11,12,13,14]. Mechanosensitive channels are widely expressed in the human body [15]. Opening of the mechanosensitive channels by mechanical signaling induces Ca+ ion influx into the cytoplasm. Increased cytoplasmic Ca+ ion concentrations can result in Ras-related C3 activation to induce cell migration [16]. Notably, mechanotransduction activates the transcription of mechanoresponsive genes depending on the activation of Yes-associated protein/transcriptional coactivator with PDZ-binding motif (YAP/TAZ). The Hippo pathway, critical for YAP/TAZ regulation, involves sequential phosphorylation of kinases, including six kinds of MAP4Ks and large tumor suppressor homolog 1/2 (LATS1/2). Since activated LATS1/2 induces the phosphorylation of YAP/TAZ and subsequent proteasome degradation of YAP/TAZ, the inactivated status of the Hippo pathway, called “Hippo off”, is positively correlated with YAP/TAZ activation [17]. The six MAP4Ks consisting of MAP4K1/HPK1, MAP4K2/GCK, MAP4K3/GLK, MAP4K4/HGK, MAP4K5/KHS, and MAP4K6/MINK are known to phosphorylate LATS1/2, and thus, they are involved in mechanosensing signaling by inhibiting YAP stabilization in various type of cells [18]. However, neither the expression nor the functions of each MAP4K in periodontal tissues have been clarified.

According to orthodontic tooth body movement applied in clinical practice, periodontal tissues, particularly PDL cells, substantially retain their mechanosensing ability. Several in vitro studies demonstrated that PDL fibroblasts (PDLF) can convert mechanical stress signals into favorable basic cellular functions, such as proliferation and differentiation, by utilizing the congenital mechanosensing mechanisms of YAP/TAZ and Rho GTPase [19]. However, if the periodontal tissue receives excess mechanical force, namely traumatic occlusion, then the tissue undergoes vacuolization and hyaline degeneration. This illustrates the difficulties that have been associated with the clinical application of mechanical force for periodontal tissue regeneration therapy. Therefore, we hypothesize that pharmacological activation of mechanosensing mechanisms in periodontal tissue, instead of mechanical force, may be a viable strategy for enhancing periodontal tissue regeneration in vivo without any deteriorating effects.

Furthermore, ligature-induced periodontitis models in murine and rattus have been utilized as models that mimic human periodontitis accompanied by severe bone defects [20,21]. The ligature-induced periodontitis models have been used to investigate the relation between periodontitis and systemic diseases and the involvement of genetic background in periodontitis onset and progression [20]. Once ligature is removed from murine and rattus molars, periodontal tissue is sequentially regenerated by an accelerated wound healing process [22]. However, the mechanisms by which accelerated regeneration are evoked in murine and rattus models remain unclear.

In the present study, we evaluated the periodontal regeneration process by measuring vertical periodontal bone regeneration and histologically focusing on cementum layer absorption and regeneration. Then, using this established murine periodontitis model, and by considering in vitro mechanistic analyses, we clarified the favorable effects of pharmacological activation of YAP/TAZ for periodontal regeneration.

## 2. Results

### 2.1. Evaluation of Sequential Alveolar Bone Recovery and Cementum Remodeling during Periodontal Tissue Regeneration

The upper second molars of 12-week-old males were ligated with 5-0 silk sutures for 14 days to induce periodontal tissue damage, and the sutures were removed and periodontal tissue was left for 0, 3, 7, 14, and 28 days and the maxillae were collected (Figure 1A). μCT analyses were conducted to analyze the distance between the CEJ and alveolar bone crest at the mesial and distal roots of the second molar on days 0, 3, 7, 14, and 28 after the 14-day ligature-induced periodontal tissue breakdown. Representative μCT images of each time point showed that sequential vertical bone growth was evident, especially in areas surrounding the mesial and distal roots of second molars (Figure 1B). The changes in the distance between the CEJ and alveolar bone crest during the 28-day regeneration period became narrower and reached 62.3 and 45.4% of the initial distance on day 14 and 28, respectively (Figure 1C). The distance in the control group was equivalent through the 28-day regeneration period, indicating no apparent aging effect during the experimental period. On day 0, it should be noted that bone resorption of all the samples equivalently reached the apex of the roots of the second molars and the sum width in the experimental group showed nominal variation between the samples (*n* = 8), indicating that the bone was equally resorbed in all the samples examined, with less technical variation if the 5-0 silk suture was ligated for 14 days. 

### 2.2. The Reappearance of TRAP-Positive Area along with Root Surface and External Root Absorption

Masson’s trichrome staining revealed loss of Sharpe’s fiber-like structures in the periodontal defects on day 0 (Figure 2A). Then, reappearance of Sharpe’s fiber-like structures were identified on days 3, 7, 14, and 28 in PDL tissue. On day 0, TRAP staining (purple color) was detected along with the root surface (blue arrowhead), and the surface of absorbed alveolar bone, but there was no apparent ALP-positive area. On day 3, even though the TRAP staining spot was identified in the deeply absorbed root dentin (red arrow), TRAP staining intensity was drastically decreased along with the root surface. In contrast, ALP intensity (brown color: green arrowhead) in PDL tissue was increased. On day 7, TRAP staining with root surface reappeared and ALP intensity in PDL was still positive. On days 14 and 28, TRAP intensity along with root surface and ALP intensity in PDL became weak. Sequential changes of TRAP intensity by quantifying the distance of TRAP-positive area along with root surface normalized against the distance of total root surface showed significant reappearance of TRAP staining intensity on day 7 (Figure 2B). Specific staining of OPN, a mediator of cementum formation, suggested that the TRAP-positive area on days 0 and 7 overlapped with the OPN-positive cementum layer, rather than dentin or PDL, indicating that cementum tissue was being metabolized. Notably, on day 28, the externally absorbed root defect observed in the early stages of regeneration remained unchanged and was filled with fibroblastic cells with OPN-positive cementum layer regeneration (red arrow).

### 2.3. Expression of Active YAP and Its Regulator LATS1/2 in Periodontal Tissue

To establish whether active YAP and its known regulator, LATS1/2, are present in periodontal tissue, the demineralized maxillae from 12-week-old males were stained with the anti-active YAP and anti-LAT1/2 antibodies, and active YAP-positive staining was detected in PDL tissue with greater intensity in the furcation area (Figure 3). Active YAP was not apparent in the epithelium of interdental papilla or gingiva. LATS1/2 was also detected in PDL tissue. Their intensities in the furcation areas overlapped with the expression pattern of active YAP. LATS1/2 was positive in the epithelium of interdental papilla and gingiva. These results implied that YAP is physiologically activated and YAP activity in PDL tissue was modulated by LATS1/2.

### 2.4. DPPA Acceleration of Periodontal Regeneration

Next, to investigate whether intravenous injection of DPPA alters the activity of YAP and PDLF proliferation, 12-week-old males were intraperitoneally injected with BrdU, housed for 24 h, and then intravenously injected with DPPA for an indicated period. Then, the demineralized maxillae from these mice were stained with anti-active YAP and anti-BrdU antibodies, and active YAP was identified after 1 h and remained positive until 4 h after the intravenous injection of DPPA (Figure 4A). BrdU positive cells were increased at 1 and 2 h and then gradually decreased.

As described in Figure 1, ligature-induced periodontitis was induced for 14 days, and periodontal tissue regeneration was assessed 14 days after the ligature removal. DPPA or DMSO was intravenously injected every other day for 14 days during the regeneration stage. Representative μCT images indicated greater vertical bone growth in the DPPA-treated group compared with the DMSO-treated group (Figure 4B). Statistical analysis used to evaluate the distance between the CEJ and alveolar bone crest revealed that this difference was significant on the ligature side (*n* = 12) (Figure 4C). Conversely, there was no apparent effect of DPPA injection on the non-ligature side. On day 14, the DPPA-treated group exhibited a wider ALP-positive area surrounding the apex and higher TRAP intensity along with the cementum layer (Figure 4D).

### 2.5. DPPA Stimulation of PDLF Proliferation Dependence on YAP/TAZ Activity

To clarify whether DPPA directly activates YAP/TAZ, PDLF were stimulated with 5 and 10 μM of DPPA for 1 h. Both 5 and 10 μM significantly induced the activation of YAP/TAZ (Figure 5A). Then, to examine the effects of DPPA stimulation on cell proliferation, PDLF were cultured with various concentrations of DPPA. Compared with the cells cultured in the absence of DPPA, the number of cells was significantly increased on day 6 in the presence of 10 and 20 μM of DPPA (Figure 5B). Next, PDLF were cultured with 10 μM of DPPA with or without verteporfin, a pharmacological inhibitor of YAP/TAZ association with the transcription factor TEAD domain (TEAD) and subsequent YAP/TAZ-TEAD complex-induced transcription (Figure 5C). The increased number of PDLF induced by DPPA was completely abrogated by the addition of verteporfin at day 6, indicating that DPPA induced PDLF proliferation is dependent on YAP/TAZ activity.

### 2.6. MAP4K4 Suppression Activates YAP

Since MAP4Ks consisting of MAP4K1/HPK1, MAP4K2/GCK, MAP4K3/GLK, MAP4K4/HGK, MAP4K5/KHS, and MAP4K6/MINK are known to directly phosphorylate LATS1/2 and induce subsequent YAP/TAZ degradation, and hence act as negative regulators of mechanosensing signaling in various type of cells, *MAP4Ks* expression in PDLF before and after the osteo/cementogenic induction was investigated by analyzing our RNA-seq data [23] (NCBI Gene Expression Omnibus: Accession Number GSE178606). According to the RPKM values of the *MAP4Ks*, *MAP4K4* was highly expressed before and after the osteogenic induction in PDLF (Figure 6A). Thus, the potential of MAP4K4 suppression for YAP/TAZ activation was examined by inhibiting endogenous *MAP4K4* expression using siRNAs specific for *MAP4K4* (Figure 6B). Among the three siRNAs examined, si-*MAP4K4*, rather than si-*MAP4K4*-2 or si-*MAP4K4*-3, efficiently suppressed MAP4K4 expression. Thus, si-*MAP4K4* was used for assays. PDLF transfected with si-*MAP4K4* expressed reduced MAP4K4 compared with the cells transfected with si-control (Figure 6C). Then, PDLF treated with control siRNA or si-*MAP4K4* were seeded with dense or sparse conditions, and the effects of MAP4K4 suppression for the activation of YAP were examined by SDS-PAGE analysis (Figure 6D). The results showed that YAP activation was significantly promoted in si-*MAP4K4*-transfected PDLF compared with control siRNA-transfected PDLF in dense conditions (Figure 6E). Then, PDLF treated with control siRNA or si-*MAP4K4* were seeded in dense conditions (Figure 6F). The expression levels of *CCN1* and *WSB2*, known YAP/TAZ target genes, were significantly increased in si-*MAP4K4*-transfected PDLF compared with control siRNA-transfected PDLF.

### 2.7. MAP4K4 Kinase Inhibition Enhances Proliferation and Osteo/Cementogenic Differentiation of PDLF by Dependently on YAP/TAZ

The expression levels of *ALPL*, *COL1A1*, and *RUNX2*, osteo/cementogenic differentiation markers, were significantly upregulated in si-*MAP4K4*-transfected PDLF compared with control siRNA-transfected PDLF (Figure 7A). Thus, PDLF were transfected with control siRNA or si-*MAP4K4* and long-term cultured in the mineralization-inducing medium for 15 days. Then, the cell proliferation rate was examined using the MTT method, and osteo/cementogenic differentiation was examined by ALP activity normalized by MTT values. The number of si-*MAP4K4*-transfected PDLF was significantly higher than that of control siRNA-transfected PDLF on days 3 and 6 (Figure 7B). The ALP actively of si-*MAP4K4*-transfected PDLF remained significantly higher than that of control siRNA-transfected PDLF throughout the entire period (Figure 7B). Consistently, the calcium deposition detected by Alizarin red S staining was increased in si-*MAP4K4*-transfected PDLF compared with control siRNA-transfected PDLF on days 9, 12, and 15 (Figure 7C).

The involvement of the kinase activity of MAP4K4 for cell proliferation and osteo/cementogenic differentiation was validated by transducing the dominant-negative MAP4K4 that was lacking the kinase domain (MAP4K4-ΔKD) into PDLF (Figure 7D). Successful generation of PDLF expressing MAP4K4-ΔKD tagged with AM-tag (PDLF-MAP4K4-ΔKD-AM) was validated by qPCR to detect transgenes and by SDS-PAGE analysis to detect transgene-delivered translational products, identified using the anti-AM antibody (Figure 7D). To further investigate whether kinase activity of MAP4K4 was required for LATS1/2 phosphorylation, cytoplasmic fractions of PDLF-empty and PDLF-MAP4K4-ΔKD-AM were loaded onto SDS-PAGE. PDLF-MAP4K4-ΔKD-AM showed significantly lower phosphorylation degree of LATS1/2 normalized by GAPDH (Figure 7E). PDLF-MAP4K4-ΔKD-AM highly expressed *ALPL* compared with PDLF-empty, but it was significantly reduced by verteporfin at 0.5 μM (Figure 7F). Next, PDLF-empty and PDLF-MAP4K4-ΔKD-AM were long-term cultured in the mineralization-inducing medium for 12 days, and the ALP activity and cell numbers were analyzed using the WST method (Figure 7G). The ALP activity in PDLF-MAP4K4-ΔKD-AM was significantly higher compared with that in PDLF-empty on days 9 and 12. Cell proliferation rate of PDLF-MAP4K4-ΔKD-AM was also higher than that of PDLF-empty. The cell number and ALP activity of PDLF-MAP4K4-ΔKD-AM treated with verteporfin at 0.25 μM decreased to those of PDLF-empty without verteporfin treatment. These results indicated that YAP/TAZ affects MAP4K4 kinase-dependent osteo/cementogenesis and cell proliferation.

## 3. Discussion

The present study demonstrated that the periodontal tissue healing process after ligature removal in ligature-induced murine periodontitis was accelerated at the early phase of the regeneration stage, the cementum layer remained after periodontal tissue breakdown, and the TRAP-positive cementum layer reappeared at the middle phase of the regeneration stage. Furthermore, by using this newly generated assessment model of periodontal tissue regeneration, pharmacological activation of YAP/TAZ by inhibition of the YAP/TAZ-negative regulatory Hippo pathway showed increased cell proliferation in PDL tissue and promoted periodontal tissue regeneration in vivo.

Cementum is a thin hard tissue found in the cervical area in human teeth, with widths of 20 to 50 μm, and consists of bone-like hardness and a similar organic/inorganic ratio [24]. However, unlike alveolar bone, cementum is not physiologically metabolized due to the nominal vascular supply and, thus, is resistant to osteoclast resorption [24,25]. Long-term bacterial infection due to chronic periodontitis destroys the cementum surface structure [24]. Notably, GTR helps reconstruct the cementum structure by initiating the formation of a blood clot and subsequent acellular cementum generation as well as connective tissue on the transiently resorbed exposed root surface [24,26]. Moreover, the cementum layer contains and releases various bioactive proteins and growth factors capable of cellular migration, proliferation, and differentiation [24]. On day3, the beginning of the periodontal regeneration stage, cementum-absorbable osteoclasts (cementoclasts) disappeared, but the OPN-positive cementum layer survived (Figure 2). This remaining cementum may be the key to periodontal tissue regeneration accompanied by a fibrous connection between cementum and alveolar bone, and alveolar vertical growth in the murine model.

A clinical case study revealed that removal of granule tissue and causative microorganisms from 2- or 3-walled vertical periodontal tissue defects by periodontal surgery without any regenerative medicine filled just around 15% of bone loss after 36 weeks [27]. In the murine periodontal regeneration model described in this study, bone loss was recovered to 62.3 and 45.4% after 14 and 28 days, respectively (Figure 1). TRAP intensity terminated and inflammatory cell infiltration was eliminated on day 3 (Figure 2). However, TRAP became positive along with the remaining cementum layer and alveolar bone surface on days 7 and 14, with stronger intensity on day 7. The reappearance of the TRAP-positive cementum layer suggests that the cementum layer is actively metabolized similarly to the alveolar bone during the regeneration period in the murine model. Furthermore, TRAP-positive layer and subsequent OPN-positive cementum regeneration were also identified at the surface of dentin resorption pit (Figure 2). Thus, addressing the differences in regeneration ability between human and mice teeth, cementum remodeling is accelerated in the murine model.

LATS1/2 and its upstream protein kinase MAP4Ks have been identified as essential molecules of the Hippo pathway [17]. Hippo off results in the YAP/TAZ nuclear localization and interaction with TEAD [28,29]. Since YAP/TAZ lacks DNA binding ability, YAP/TAZ exerts its transcriptional potential by binding with TEAD to guide TEAD into transcription regulatory sites in the open chromatin region of YAP/TAZ target gene loci [29,30]. Nuclear localized active YAP was broadly expressed in PDL tissue, with higher intensity in furcation areas (Figure 3). Since this expression pattern in PDL tissue was similar to that of YAP/TAZ-negative regulators, such as LATS1/2, YAP/TAZ signal activity is physiologically regulated by LATS1/2 in PDL tissue. DPPA, a phosphatidic acid, was previously identified as a potent accelerator of YAP/TAZ signaling by LATS1/2 kinase inhibition in breast cancer cells [31]. DPPA intravenous administration enhanced alveolar bone regeneration (Figure 4). The ALP-positive area tended to increase in the regenerated periodontal tissue of the DPPA-treated group. However, PDLF stimulated with DPPA did not show higher ALP activity compared with non-stimulated cells in vitro (data not shown). These results suggested that YAP/TAZ activation by DPPA induced periodontal tissue regeneration mainly by enhancing cell proliferation of PDL stem cells, which eventually differentiated into ALP-positive cells in the PDL. YAP/TAZ and the TEAD complex are known to directly associate with the gene regulatory domain and evoke osteogenic gene expression in osteoblasts and mesenchymal stem cells [32]. Epigenetic alteration of chromatin accessibility by genome-wide modulators, such as histone acetyltransferases and histone methyltransferase, and local modulators, such as NFkBiζ, are deeply linked with hard tissue formation and inflammation-induced alveolar bone destruction [33,34,35]. In contrast to osteogenic genes that were directly transcribed by YAP/TAZ transcriptional ability, YAP/TAZ involvement in cell proliferation is linked with metabolic reprogramming [36]. In YAP/TAZ activating cells, YAP/TAZ induced the expression of the rate-limiting enzyme, ornithine decarboxylase 1, which increased the polyamine synthesis. Increased polyamine levels enhanced the hypusination of eukaryotic translation factor 5A, which is required to translate histone demethylase LSD1. Whole genomic alteration of chromatin accessibility by LSD1, a transcriptional repressor, oriented the cells for the proliferative phase.

Endogenously, DPPA is produced by three common metabolic pathways: phospholipase D (PLD) hydrolyzes phosphatidylcholine, lysophosphatidic acid acyltransferase (LPAAT) acylates lysophosphatidic acid, and diacylglycerol kinase (DGK) phosphorylates diacylglycerol to produce DPPA [36]. RNA-seq data of PDLF before and after 6-day osteo/cementogenic differentiation revealed that *PLD1* coding PLD and *PDKA*, *PDKH*, *PDKK*, and *PDKZ* coding PDK family members PDKA, PDKH, PDKK, and PDKZ, were abundantly expressed, and their expressions were not altered during 6-day induction (NCBI Gene Expression Omnibus: Accession Number GSE178606) [23]. Since the expression levels of *AGPAT1* coding LPAAT and *DGKI* coding DGKI increased 5.7-fold and 17.7-fold after osteo/cementogenic induction, LPAAT and DGKI may play pivotal roles in maintaining periodontal tissue homeostasis by increasing DPPA levels in vivo.

MST1/2 is considered the protein kinase of LATS1/2, and more recently, MAP4Ks have been regarded as an alternative upstream protein kinase of LATS1/2 [37,38]. Among six MAP4Ks, MAP4K4 was highly expressed in PDLF (Figure 6A). Since the MST1/2, LATS1/2, and MAP4Ks was more highly activated when cells were cultured in dense conditions [17,38], the effects of MAP4K4 suppression for YAP activation were significant in dense conditions but not in sparse conditions in PDLF (Figure 6D–F). Truncated MAP4K4 lacking the kinase domain receives the upstream signal, but is unable to phosphorylate LAT1/2, and thus, acts as dominant-negative MAP4K4 (Figure 7) [17]. Overexpression of dominant-negative MAP4K4 increased ALP activity and cell proliferation, suggesting that kinase activity was critical for MAP4K4-induced cell proliferation and osteo/cementogenic differentiation of PDLFs. Similar to DPPA functions (Figure 5), the suppression of MAP4K4 increased cell proliferation of PDLFs; however, MAP4K4 suppression, but not DPPA stimulation, induced osteo/cementogenic differentiation in vitro. This discrepancy may arise from the existence of other undefined MAP4K4 downstream pathways, which negatively regulate osteo/cementogenic differentiation in PDLFs. Moreover, MAP4K4 is one of the members of the MAP4K family, and the sequential kinase of MAP4K-MAP3K-MAPKK-MAPK participates in various cellular functions. Particularly, MAPK signals to JNK are usually associated with osteogenic suppression [39,40]. Further studies are required to evaluate the usefulness of MAP4K4 inhibition for periodontal regeneration in vivo and to identify specific MAP4K4 downstream target signals that contribute to promoting osteo/cementogenic differentiation when MAP4K4 is suppressed.

Systemic DPPA administration was chosen in this study due to its low solubility in aqueous solutions. Even though intravenous injection of DPPA did not show any visible adverse effects, the oncogenic properties of YAP/TAZ may interrupt clinical use as a systemic drug. Therefore, future studies are necessary to improve the water solubility and half-life of DPPA and to develop water-soluble inhibitors of LATS1/2 and MAP4K4 to overcome the current matter of topical application.

## 4. Material and Methods

### 4.1. Reagents

We purchased 16:0 PA 1,2-dipalmitoyl-sn-glycero-3-phosphate (DPPA) from Avanti Polar Lipids (Birmingham, AL, USA).

### 4.2. Experimental Animals

All experimental procedures conformed to the “Regulations for Animal Experiments and Related Activities at Tohoku University” and were reviewed by the Institutional Laboratory Animal Care and Use Committee of Tohoku University and approved by the President of the University. (Permit No. 2018DnA-043-08). Eleven-week-old male C57BL6/J mice (specific pathogen-free grade) were purchased from CLEA Japan, Inc. (Tokyo, Japan). The mice were randomly allocated, and five mice were housed per cage. The mice were fed standard rodent chow and water ad libitum, and they were given 1 week to adapt to the new environment before the experiment. The mice were anesthetized, and silk ligatures (Elp Sterile Blade Silk, Black, 5–0, Akiyama Medical MFG. Co., Ltd., Tokyo, Japan) were tied around their second maxillary molars for 14 days. After the 14-day periodontal inflammation period, the silk suture was removed and the periodontal tissue was left for 3, 7, 14, and 28 days. For evaluating the effects of DPPA on periodontal regeneration, DPPA dissolved in dimethyl sulfoxide (DMSO) was intravenously administered (3 mg/kg) in the tail vein every other day. Mice in the control group were administered the same amount of DMSO in the same manner and frequency.

### 4.3. Micro-Computed Tomography

Micro-computed tomography (μCT) was conducted as described previously [33]. The hemimaxillae were dissected, fixed for 24 h in 4% paraformaldehyde, and stored in Dulbecco’s phosphate-buffered saline (DPBS) at 4 °C. Samples were scanned using a μCT scanner (Scanxmate-E090, Comscantecno Co., Ltd., Yokohama, Japan) with an isotropic resolution of 50 µm. X-ray images were recorded for 360° rotation of samples at 80 kV/83 µA using Xsy FP software (version 2.1, Comscantecno Co., Ltd., Yokohama, Japan). Three-dimensional images were reconstructed in coneCTexpressI software (version 1.59, WhiteRabbit Co., Ltd., Tokyo, Japan) and TRI/3D-BON software (version R2.00.06.0-H-64, Ratoc System Engineering Co., Ltd., Tokyo, Japan). Images were taken for analysis at designated positions.

All images were re-oriented with the tomographic coronal plane of the second molar 2D images made parallel to the buccal-lingual center line and coronal-apical center line. The vertical distances from the cemento-enamel junction (CEJ) to the alveolar bone crest at the mesial and distal roots were measured and summed. This sum was used for quantitatively comparing bone regeneration levels in the periodontal regeneration stage.

### 4.4. Histology

Immunostaining, Masson’s trichrome staining, and tartrate-resistant acid phosphatase (TRAP)/alkaline phosphatase (ALP) double staining were performed on 5 µm-thick paraffin sections. Mice were perfused with DPBS to remove circulating blood and then with 4% paraformaldehyde in PBS. The maxillae were removed and fixed in 4% paraformaldehyde in DPBS at 4 °C for 24 h. They were then decalcified with 0.134 mol of ethylenediaminetetraacetic acid in DPBS at 4 °C and dehydrated by passing through a graded ethanol series, placed in xylene, and embedded in paraffin. For immunohistochemical staining, the sections were dewaxed and stained using a MACH4 Universal HRP-Polymer Detection kit (BRR4012: Biocare Medical, Pacheco, CA, USA) according to the manufacturer’s instructions. Sections were incubated with rabbit polyclonal anti-active YAP (EPR19812: Abcam, Cambridge, UK, 1:30), anti-LATS1/2 (20276-1-AP: Proteintech, Rosemont, IL, USA, 1:500) or osteopontin (OPN; LF-123: kind gift from Dr. Larry Fisher, NIDCR, 1:2000), overnight at 4 °C. The immune complexes were formed using MACH4 HRP-Polymer for 20 min at room temperature, and the peroxidase reaction in the immune complexes was visualized by a chromogen substrate 3,3’-diaminobenzidine reaction. Then, the sections were counterstained with hematoxylin and mounted. The dewaxed sections were used for Masson’s Trichrome staining. The sections were also stained using a TRAP/ALP stain kit (FUJIFILM Wako Pure Chemical Corporation, Osaka, Japan) according to the manufacturer’s instructions to sequentially detect TRAP and ALP-positive cells. Histological images were captured using an upright microscope (DM6000 B: Leica, Wetzlar, Germany) with a digital camera (DP28: Olympus, Tokyo, Japan).

### 4.5. Cell Culture and Stable Cell Generation

Human PDLF were purchased from Lonza Inc. (Walkersville, MD, USA). PDLF were maintained in low glucose Dulbecco’s Modified Eagle Medium (DMEM; Thermo Fisher Scientific, Carlsbad, CA, USA) supplemented with 100 units/mL of penicillin, 100 μg/mL of streptomycin, and 10% fetal bovine system. PDLFs were cultivated at 37 °C under humidified atmospheric conditions (5% CO_2_ and 95% air). For osteogenic induction, PDLF were cultured in an induction medium (low glucose DMEM with ascorbic acid (100 μg/mL) and β-glycerophosphate (10 mM)). The medium was replaced every 3 days. For generating PDLF stably expressing kinase domain-deleted MAP4K4 (MAP4K4-ΔKD), which lacks the kinase domain (M^1^-I^289^), tagged with AM-tag (MAP4K4-ΔKD-AM), MAP4K4-ΔKD-AM were amplified from the human MAP4K4 cDNA sequence (Clone ID100062694), which was purchased from DNAFORM (Kanagawa, Japan), and the AM-tag sequence was amplified from the AM-tag vector (pAM_1C Empty Vector, Active Motif, Carlsbad, CA). The amplified MAP4K4-ΔKD and AM sequences were cloned into pBluescript II sk(+) in-frame to generate MAP4K4-ΔKD-AM, and the entire sequence were verified. Then, the amplified MAP4K4-ΔKD-AM sequence were ligated into pLVSIN-CMV-Pur Vector (Takara Bio Inc., Otsu, Japan) to obtain pLVSIN-CMV-Pur-MAP4K4-ΔKD-AM and the sequence was verified again. Next, pLVSIN-CMV-Pur-MAP4K4-ΔKD-AM or pLVSIN-CMV-Pur-empty and lentivirus packaging plasmid (Lentiviral High Titer Packaging Mix, Takara Bio Inc.) were co-transfected into G3Thi cells, a subtype of 293T cells (Takara Bio Inc.), using X-tremeGene (Roche, Basel, Switzerland) for 24 h. Then, the medium was replaced with a serum-free medium and cultured for 24 h, and the supernatants containing lentivirus particles were collected and concentrated using Lenti-X™ Concentrator (Takara Bio Inc.). Next, PDLF were infected with an equal amount of pLVSIN-CMV-Pur-MAP4K4-ΔKD-AM or pLVSIN-CMV-Pur-empty lentivirus with 8 μg/mL of polybrene (H9268, Sigma-Aldrich, St. Louis, MO, USA) in medium containing 20% fetal bovine system and cultured for 24 h. Then, the medium was replaced with a growth medium, and the cells were further cultured for 24 h for cell recovery. Then, PDLF were selected using 0.5 μg/mL of puromycin for 12 days. The medium was replaced every 3 days. After 12 days of selection, PDLF overexpressing MAP4K4-ΔKD-AM (PDLF-MAP4K4-ΔKD-AM) and control (PDLF-empty) cells were expanded and used for further assessment.

### 4.6. Quantitative PCR (qPCR) Analysis

Total RNA from PDLF was purified using RNAiso plus (Takara Bio Inc., Otsu, Japan). Using the ReverTra Ace qPCR RT master mix with gDNA remover (Toyobo Life Science, Tokyo, Japan), 0.5 μg of total RNA was reverse transcribed into cDNA using mixed primers. qPCR reactions were prepared using the KAPA SYBR Fast qPCR kit (KAPA BIOSYSTEMS, Woburn, MA, USA). Human *HPRT* was used as an internal reference control. PCR primer sequences for target genes are shown in Table A1.

### 4.7. Transient Transfection of siRNA

RNA sequences for targeting MAP4K4 by siRNA were selected using Enhanced siDi-rect, a web-based target-specific siRNA design software. Control siRNA was previously described [23,41]. siRNAs were generated by Sigma-Aldrich. The siRNA sequences of three kinds of siRNAs for *MAP4K4* (si-MAP4K4), and that of control siRNA (si-control), are shown in Table A2. siRNAs were forward-transfected into PDLF at a final concentration of 10 nM using Lipofectamine RNAiMAX reagent (Thermo Fisher Scientific) and then incubated for 24 h.

### 4.8. Immunoblotting

Reduced samples were loaded onto NuPAGE Bis-Tris (Thermo Fisher Scientific) gels in MOPS buffer, and separated proteins were transferred onto a polyvinylidene fluoride membrane for immunodetection with the anti-MAP4K4 (GeneTex Inc., Irvine, CA USA, 1:2000), active YAP (EPR19812: Abcam), anti-AM (AbFlex^®^ #91111: Active Motif, Carlsbad, CA, USA, 1:2000), anti-LATS1/2 (20276-1-AP: Proteintech, Rosemont, IL, USA), anti-phospho-LATS1/2 (bs-7913R: Bioss Antibodies, Woburn, USA), and GAPDH (GTX100118: GeneTex Inc., 1:1000) antibodies as the primary antibodies, and the HRP-conjugated goat anti-rabbit IgG (#7074: Cell Signaling Technologies, 1:2000) or HRP-conjugated goat anti-mouse IgG (#91196: Cell Signaling Technologies, 1:2000) antibodies as the secondary antibodies.

### 4.9. ALP Activity

ALP activity was measured as previously described [42]. It was normalized to cell numbers from a parallel cell culture that was quantified using the WST Cell Count Kit-8 (341-08001: Dojindo, Kumamoto, Japan) or the MTT Cell Count Kit (23506-80: Nacalai Tesque, Kyoto, Japan).

### 4.10. Alizarin Red S Staining

PDLF were washed twice with DPBS and fixed with 70% ethanol for 10 min. Fixed cells were washed with distilled H_2_O and stained with 1% Alizarin red S (pH 4.2) solution.

### 4.11. Statistical Analysis

Statistical analysis was performed by one-way analysis of variance, followed by the Bonferroni test (Figure 2B, Figure 5B,C and Figure 7F,G) and two-tailed unpaired Student’s *t*-tests (Figure 1, Figure 4, Figure 5A, Figure 6 and Figure 7B,E).

## 5. Conclusions

The present results demonstrated for the first time that the active cementum remodeling in the periodontal tissue regeneration period was necessary for mice periodontal tissue to exhibit aggressive regeneration tropism and YAP/TAZ activation induced by inhibiting upstream negative regulators, such as LATS1/2 and MAP4K4, promoted periodontal regeneration by enhancing cell proliferation. Therefore, pharmacological activation of YAP/TAZ activation has therapeutic potential for accelerating periodontal tissue regeneration.

## Figures and Tables

**Figure 1 ijms-24-00970-f001:**
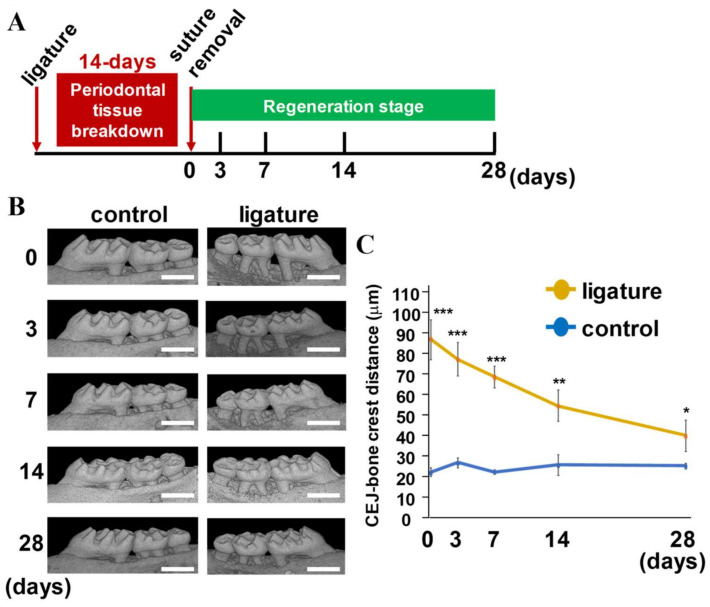
Quantitative analyses of the murine periodontal tissue regeneration model. (**A**,**B**) Schematic view of experimental design of the murine periodontal regeneration model and representative μCT images of maxilla used for μCT analyses. (**C**) The vertical distances from the CEJ to the alveolar bone crest at the mesial and distal roots on days 0, 3, 7, 14, and 28 were measured and summed (*n* = 5). * *p* < 0.05; ** *p* < 0.01; *** *p* < 0.001 significantly different from non-ligature side. Scale bars correspond to 1000 μm.

**Figure 2 ijms-24-00970-f002:**
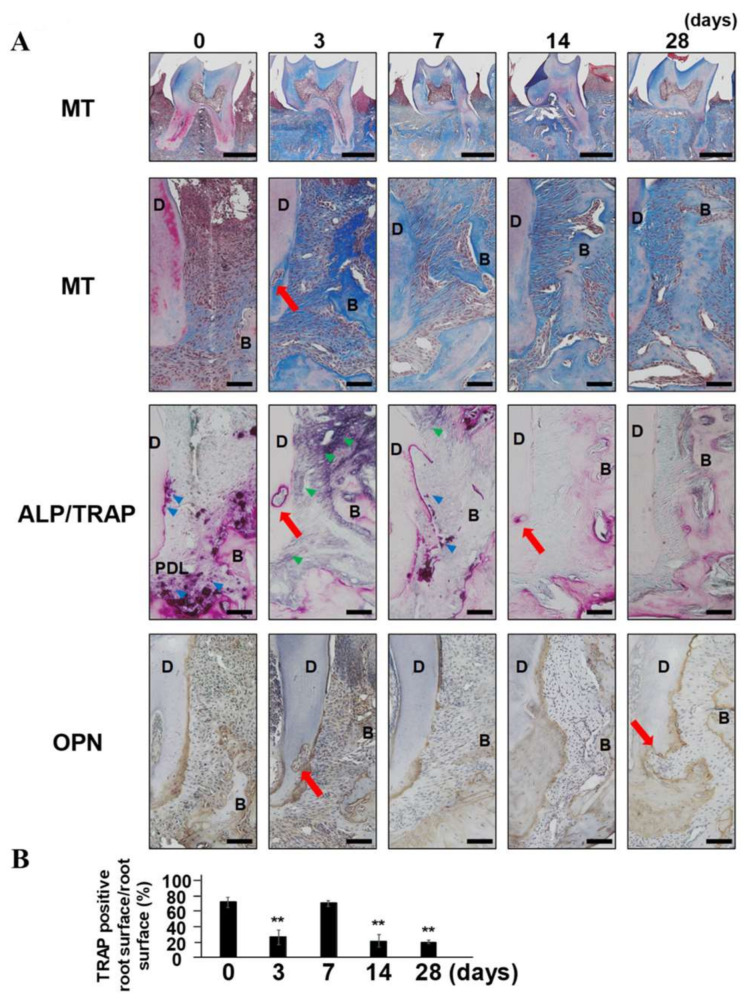
Cementum remodeling and external root resorption in the murine periodontal tissue regeneration model. (**A**) Demineralized male maxilla sections collected on days 0, 3, 7, 14, and 28 were stained with TRAP/ALP double staining, Masson’s trichrome, and α-OPN. Scale bars correspond to 500 and 100 μm at low and high magnification, respectively. (**B**) The distance of TRAP-positive area along with root surface was normalized by total distance of root surface (*n* = 3) and the ratio was comparatively evaluated. ** *p* < 0.01 significantly different from day 0. MT = Masson’s trichrome staining. PDL = periodontal ligament tissue, C = cementum, D = dentin, B = bone, P = pulp. Red arrows indicate an external dentin-absorbed area.

**Figure 3 ijms-24-00970-f003:**
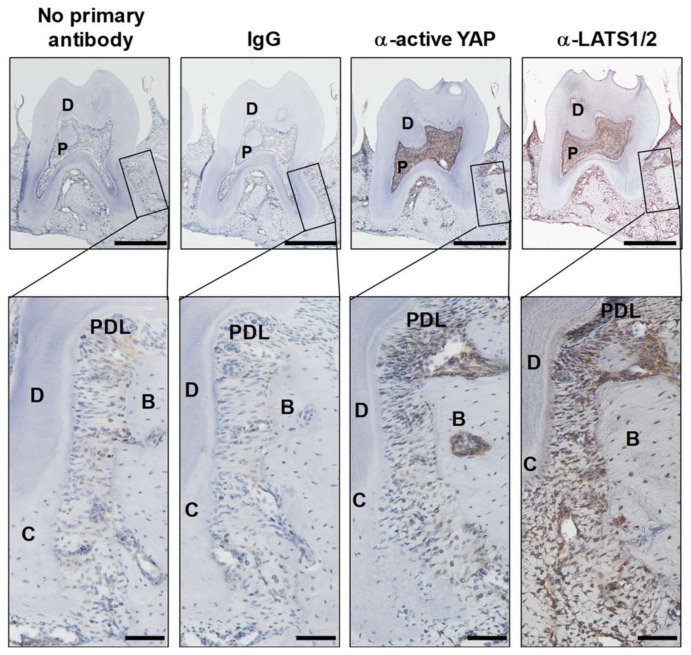
Active YAP and LATS1/2 expression in periodontal ligament tissue. Demineralized 3-month-old male maxilla sections were stained with or without rabbit IgG, or α-active YAP and α-LATS/12. Scale bars correspond to 500 and 100 μm at low and high magnification, respectively. PDL = periodontal ligament tissue, C = cementum, D = dentin, B = bone, P = pulp.

**Figure 4 ijms-24-00970-f004:**
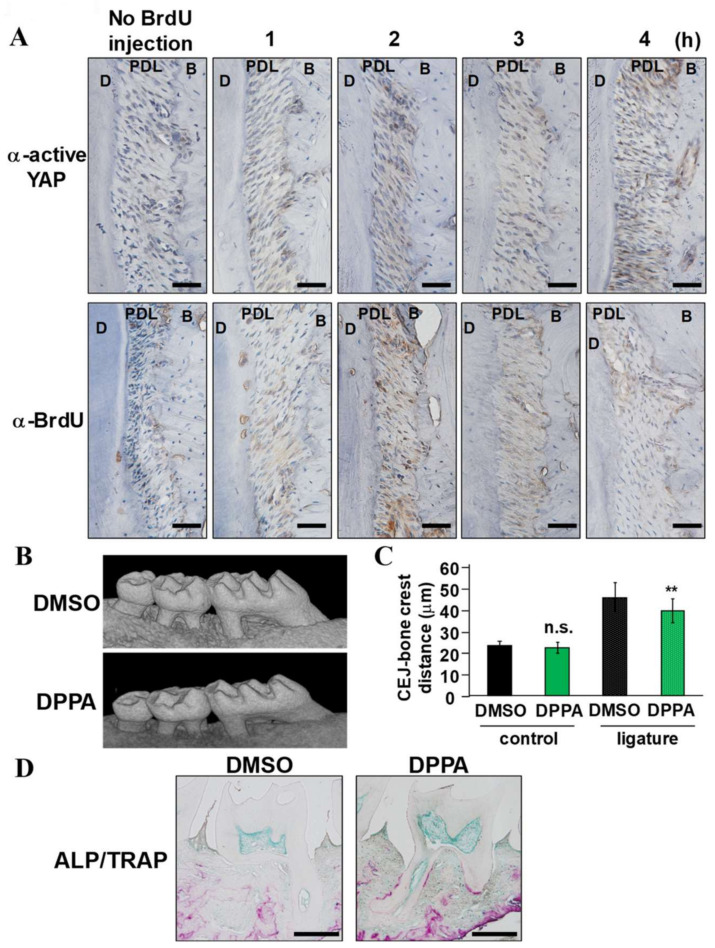
**DPPA administration during the regenerative period promotes alveolar bone regeneration.** (**A**) Twelve-week-old males were intraperitoneally injected with BrdU, housed for 24 h, and then intravenously injected with DPPA for the indicated time. Then, the demineralized sections were stained with α-active YAP and α-BrdU. (**B**) Representative μCT images of maxilla used for μCT analyses and (**C**) the vertical distances from the CEJ to the alveolar bone crest at the mesial and distal roots on day 14 (*n* = 12). (**D**) Demineralized male maxilla sections collected on day 14 were stained with TRAP/ALP double staining. Scale bars correspond to 500 μm. ** *p* < 0.01 significantly different from DMSO-treated group. n.s. = not significant, MT = Masson’s trichrome staining, PDL = periodontal ligament tissue, C = cementum, D = dentin, B = bone, P = pulp.

**Figure 5 ijms-24-00970-f005:**
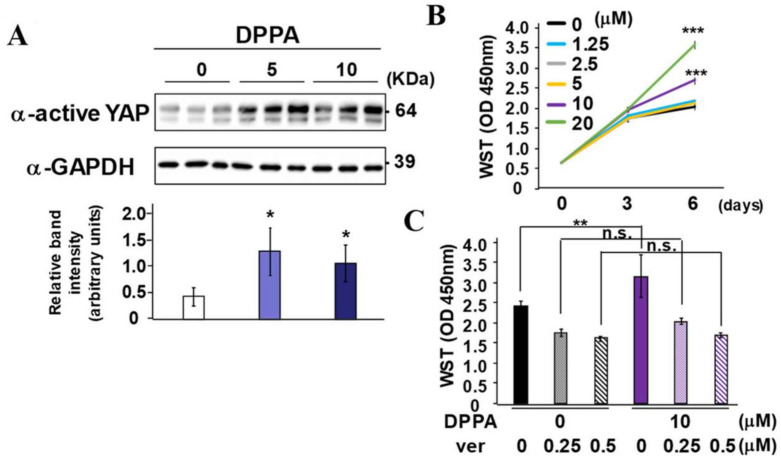
DPPA activates YAP and promotes cell proliferation in vitro. (**A**) PDLF were cultured in the presence of DPPA for 24 h, and the YAP activation was analyzed by SDS-PAGE using GAPDH as the loading control. (**B**) PDLF were cultured in various concentrations (0, 1.25, 2.5, 5, 10, and 20 μM) of DPPA for 6 days, and the number of cells was counted. (**C**) PDLF were cultured with DPPA (10 μM) in the presence of verteporfin (0, 0.25, and 0.5 μM) for 6 days, and the number of cells was counted. * *p* < 0.05; ** *p* < 0.01; *** *p* < 0.001 significantly higher than the cells in the absence of DPPA (**B**) and lower than the cells in the absence of verteporfin (**C**). n.s. = not significant. ver = verteporfin.

**Figure 6 ijms-24-00970-f006:**
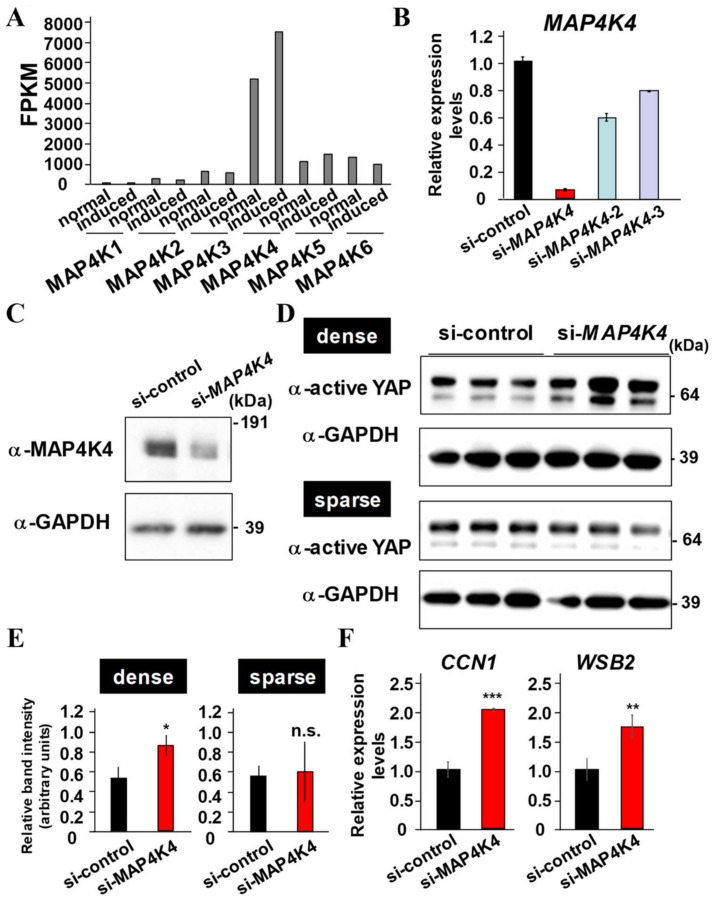
Identification of MAP4K4 as principle MAP4Ks for YAP activation in PDLF. (**A**) Six kinds of *MAP4K4s* coding gene expression in PDLF. Processing of RNA-seq data reveals *MAP4K4* was highly expressed before and after the osteo/cementogenic induction compared with other *MAP4Ks* in PDLF. (**B**) Three independent siRNAs for *MAP4K4* were transfected into PLDF and total RNA was collected to quantify the expression of *MAP4K4*. *HPRT* was used for normalization. (**C**) PDLF were transfected with si-control or si-*MAP4K4* for 24 h, and whole cell lysate was collected and MAP4K4 expression was examined using specific antibodies. The membrane was also incubated with the antibody for GAPDH as a loading control. (**D**,**E**) PDLF were transfected with si-control or si-*MAP4K4* for 24 h and then re-seed with dense (1.2 × 10^6^/well in a 6-well plate) or sparse (1.5 × 10^5^/well in a 6-well plate) density and then cultured for another 24 h. Whole cell lysate was collected and the activation of YAP was examined using a specific antibody (**D**). Band intensities are normalized against those obtained using the antibody for GAPDH (**E**). (**F**) PDLF were transfected with si-control or si-*MAP4K4* for 24 h and then re-seed with high density and then cultured for another 24 h. Then, total RNA was collected to quantify the expression of *CCN1* and *WSB2*. *HPRT* was used for normalization. * *p* < 0.05; ** *p* < 0.01; *** *p* < 0.001 significantly higher than the cells transfected with si-control. n.s. = not significant.

**Figure 7 ijms-24-00970-f007:**
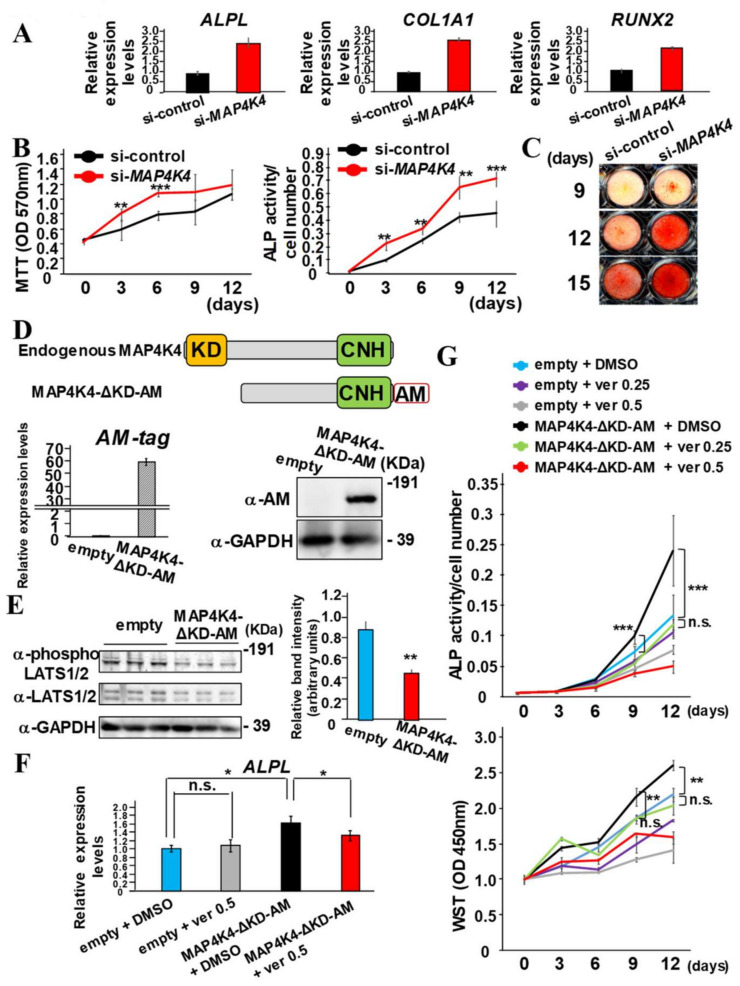
MAP4K4 kinase inhibition promotes cell proliferation and osteo/cementogenic differentiation of PDLF. (**A**) PDLF were transfected with si-control or si-*MAP4K4* for 24 h and then re-seed with high density and then cultured for another 24 h. Then, total RNA was collected to quantify the expression of *ALPL*, *COL1A1*, and *RUNX2*. *HPRT* was used for normalization. (**B**) PDLF were transfected with si-control or si-*MAP4K4* for 24 h and cultured in the mineralization-inducing medium. Cell numbers were counted by MTT assay, and ALP activities were measured every 3 days. ALP activities were normalized by WST values. (**C**) Alizarin red S staining was performed on days 9, 12, and 15. (**D**) Schematic view of functional domains of endogenous MAP4K4 and MAP4K4-ΔKD-AM. Total RNA was collected from PDLF-empty and PDLF-MAP4K4-ΔKD-AM to quantify transgene expression by using AM-tag specific primer pairs. *HPRT* was used for normalization. Whole cell lysate was collected from PDLF-empty and PDLF-MAP4K4-ΔKD-AM, and the expression of MAP4K4-ΔKD-AM proteins were examined using AM-tag specific antibodies. The membrane was incubated with the antibody for GAPDH as a loading control. (**E**) Cytoplasmic fractions were collected from PDLF-empty and PDLF-MAP4K4-ΔKD-AM, and the expression of phosphorylated LATS1/2 and total LATS1/2 proteins was examined using specific antibodies. The membrane was incubated with the antibody for GAPDH as a loading control. (**F**) PDLF-empty and PDLF-MAP4K4-ΔKD-AM were cultured in the presence of verteporfin (0.5 μM) for 24 h and the *ALPL* expression was quantified. *HPRT* was used for normalization. (**G**) PDLF-empty and PDLF-MAP4K4-ΔKD-AM were cultured in the mineralization-inducing medium for 12 days. Cell numbers were counted by WST assay, and ALP activities were measured every 3 days. KD = kinase domain, CNH = citron homology domain, ver = verteporfin. * *p* < 0.05; ** *p* < 0.01; *** *p* < 0.001 significantly higher from the cells transfected with si-control (**B**) and significantly different from PDLF-empty treated with DMSO (**F**).

## Data Availability

RNA-seq data (NCBI Gene Expression Omnibus: Accession Number GSE178606).

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
