# Peer review of "Pharmacological Activation of YAP/TAZ by Targeting LATS1/2 Enhances Periodontal Tissue Regeneration in a Murine Model"

_ijms, 2023, doi:10.3390/ijms24020970_

Round 1
Reviewer 1 Report
Manuscript “Pharmacological activation of YAP/TAZ by targeting LATS1/2 enhances alveolar bone regeneration in a murine model” represents a contribution to field of molecular sciences.
Text is clear and easy to read. Conclusions are consistent with the evidence and arguments presented. The research topic is original. The results presented are a contribution not only to fundamental science but also to applied science.
Before accepting the manuscript, it is essential that the authors:
In accordance with the objective of the research:
“In the present study, we evaluated the periodontal regeneration process by measuring vertical periodontal bone regeneration and histologically focusing on cementum layer absorption and regeneration. Then, using this established murine periodontitis model, and by considering in vitro mechanistic analyses, we clarified the favorable effects of pharmacological activation of YAP/TAZ for periodontal regeneration.”
It is necessary to add a special chapter titled Conclusion at the end of the manuscript. In conclusion, explain in detail what was achieved with the results presented.
Author Response
We wish to express our appreciation to the first reviewer for their insightful comments on our paper. The comments have helped us significantly improve the paper.
Reviewer #1: Text is clear and easy to read. Conclusions are consistent with the evidence and arguments presented. The research topic is original. The results presented are a contribution not only to fundamental science but also to applied science. Before accepting the manuscript, it is essential that the authors:In accordance with the objective of the research:“In the present study, we evaluated the periodontal regeneration process by measuring vertical periodontal bone regeneration and histologically focusing on cementum layer absorption and regeneration. Then, using this established murine periodontitis model, and by considering in vitro mechanistic analyses, we clarified the favorable effects of pharmacological activation of YAP/TAZ for periodontal regeneration.” It is necessary to add a special chapter titled Conclusion at the end of the manuscript. In conclusion, explain in detail what was achieved with the results presented.
We strongly appreciate the first reviewer's comment on this point. As the reviewer suggested, we newly generated `Conclusion` part at the end of the manuscript and described what we have done in this study as follow.
" The present results demonstrated for the first time that the active cementum remodeling in the periodontal tissue regeneration period was necessary for mice periodontal tissue to exhibit aggressive regeneration tropism and YAP/TAZ activation induced by inhibiting upstream negative regulators, such as LATS1/2 and MAP4K4, promoted periodontal regeneration by enhancing cell proliferation. Therefore, pharmacological activation of YAP/TAZ activation has therapeutic potential for accelerating periodontal tissue regeneration. "
Accordingly, the 1st paragraph of `Discussion` part was revised to avoid sentence duplication.
We hope that the first reviewer approves of this revised version.
Reviewer 2 Report
This paper is well written. The science is sound too. I only have a few minor suggestions.
1. I think it would be helpful if the author could include the anti-active and anti-LATS1/2 dilution.
2. “The upper second molars of 12-week-old males were ligated with 5-0 silk sutures for 14 days to induce periodontal tissue inflammation and damage.” Do you have any evidence for the presence of inflammation? “Masson’s trichrome staining revealed inflammatory cell infiltration into the periodontal defects on day 3 (Fig. 2).” Please explain how do you use this staining to identify inflammatory cell infiltration?
3. “TRAP staining intensity was drastically 214 decreased along with the root surface.” Is that possible to quantify the TRAP staining? Please also explain bone staining differences among the groups as some were stained as dark blue color and some have light color. Thanks.
4. Figure4D. It would be great if you could use arrow to indicate ALP-positive and TRAP-positive cells?
Author Response
We wish to express our appreciation to the second reviewer for their insightful comments on our paper. The comments have helped us significantly improve the paper.
- I think it would be helpful if the author could include the anti-active and anti-LATS1/2 dilution.
Thank you for your suggestion. The concentrations of anti-active YAP and anti-LATS1/2 are 1:30 and 1:500, respectively.
Accordingly, we revised `Histology` part.
- The upper second molars of 12-week-old males were ligated with 5-0 silk sutures for 14 days to induce periodontal tissue inflammation and damage.” Do you have any evidence for the presence of inflammation? “Masson’s trichrome staining revealed inflammatory cell infiltration into the periodontal defects on day 3 (Fig. 2).” Please explain how do you use this staining to identify inflammatory cell infiltration?
Thank you for your comments. As the reviewer suggests Masson’s trichrome staining can help us distinguishing cell-rich area from collagenous extracellular matrix but unable to help us identifying the existence of inflammation or cell types. We therefore revised the 1st sentence of 1st paragraph of `Results` part as follow.
" The upper second molars of 12-week-old males were ligated with 5-0 silk sutures for 14 days to induce periodontal tissue damage, and the sutures were removed and periodontal tissue was left for 0, 3, 7, 14, and 28 days and the maxillae were collected (Fig. 1A). "
We also revised the 1st and 2nd sentences of 2nd paragraph of `Results` part as follow.
"Masson’s trichrome staining revealed loss of Sharpe’s fiber-like structures in the periodontal defects on day 0 (Fig. 2A). Then, reappearance of Sharpe’s fiber-like structures were identified on days 3, 7, 14, and 28 in PDL tissue. "
- TRAP staining intensity was drastically 214 decreased along with the root surface.” Is that possible to quantify the TRAP staining? Please also explain bone staining differences among the groups as some were stained as dark blue color and some have light color. Thanks.
Thank you for your suggestion. As the reviewer suggests, we quantified the distance of TRAP-positive root surface and normalized against total distance of root surface in each section. There was significant decline at day 3 but it was recovered at day 7 as shown in newly generated Fig. 2B in the revised manuscript.
Accordingly, we inserted explanatory sentences of Fig. 2B in `Results` part as follow.
"Sequential changes of TRAP intensity by quantifying the distance of TRAP-positive area along with root surface normalized against the distance of total root surface showed significant reappearance of TRAP staining intensity on day 7 (Fig. 2B). "
Accordingly, we revised figure legend of Fig. 2B as follow.
"The distance of TRAP-positive area along with root surface was normalized by total distance of root surface (n = 3) and the ratio was comparatively evaluated. **p < 0.01 significantly different from day 0. "
Thank you for your comment about the bone straining difference among the group. As you suggest, blue staining intensity at day 3 is relatively higher than that at other time points. Generally, the blue staining density of Aniline blue (Masson’s trichrome) stain is not stoichiometrically linked with collagen content. As Rieppo et al reported that Masson’s trichrome can be used for quantifying collagen content only after proteoglycan degradation by papain treatment in articular cartilage (https://journals.plos.org/plosone/article?id=10.1371/journal.pone.0224839). Therefore, we think that the different blue intensity may arise from the different expression/accumulation level of proteoglycans and their glycosaminoglycan chains, and glycoproteins in alveolar bone during regeneration period. However, this is still too speculative to describe in the revised manuscript. As next research topic, we would like to analyze sequential changes of expression level of bone-related proteoglycans such as DMP-1, lumican, decorin, biglycan, and fibromodulin to seek their functions in alveolar bone regeneration such as collagen fibrillogenesis and subsequent mineralization initiation.
- It would be great if you could use arrow to indicate ALP-positive and TRAP-positive cells?
Thank you for your suggestion. As the reviewer suggested we indicated ALP-positive cells in PDL tissue with green arrowhead and TRAP-positive cells in PDL tissue with blue arrowhead in revised Figure 2.
Accordingly, we revised the explanatory sentences in `Result` part.
In the revised manuscript, we addressed all of your helpful comments as described above. We hope that second reviewer approves of this revised version.